# Influence of Climatic Variables on the Stem Growth Rate in Trees of a Tropical Wet Forest

**Juan Carlos Valverde [1],\*** , **Dagoberto Arias-Aguilar [2]** , **Marvin Castillo-Ugalde [2]** **and Nelson Zamora-Villalobos [2]**

1 Facultad de Ciencias Forestales, Universidad de Concepción, Victoria 631, Región del Bío-Bío, Concepcion 4030000, Chile
2 Forest Engineering School, Tecnológico de Costa Rica, Cartago 302101, Costa Rica; darias@tec.ac.cr (D.A.-A.)
\* Correspondence: juvalverdeo@udec.cl

**Abstract:** The growth of tropical wet forests has a significant relationship with the climate; aspects such as temperature and precipitation affect the species; however, few studies have characterized the stem growth rate of tropical tree species. This study's objective was to characterize the effects of climatic variation on the interannual stem growth rate of eight species in tropical wet forest. Six trees per species were selected ($n = 48$ trees), and a dendrometer was installed to measure diametric growth bi-monthly between 2015 and 2018 (3 years), complemented with environmental measurements, to determine their growth equations from environmental variables and, finally, to define the relationship between the wood density and the stem growth rate. The results showed an average stem growth from 0.45 to 4.35 mm year$^{-1}$, and 40 to 70% growth occurred in the months with the highest rainfall. Also, species with higher wood densities were found to have lower stem growth rates. Finally, the analysis of stem growth rate showed a significant relationship in all species between the variables of temperature and precipitation ($R^2$ adj 0.88 to 0.96). Our results suggest that species with greater stem growth rates in wet tropical forests are more susceptible to climate changes, which may affect their dynamics in the face of potential drought scenarios and heat waves associated with climate change.

**Keywords:** dendrometers; stem growth; climate change; Costa Rica

## 1. Introduction

Tree growth and regional or biome-level forest productivity are directly and indirectly related to climatic conditions [1,2]. The variation in temperature, relative humidity, and precipitation strongly affects the availability of resources (e.g., light, water, and nutrients) for individuals' survival, growth, and development [3,4]. Therefore, species have developed multiple morphological, physiological, and biochemical mechanisms to adapt to changes in climate and minimize potential stress events [5,6].

Stem growth rate has been considered a trait of interest in evaluating tree response to climatic conditions [3,7]. Dendrochronological studies have shown a strong relationship between stem growth rate and climatic conditions in boreal, temperate, and tropical forests [8,9]. For example, trees decrease their growth rate according to the level of environmental resource restriction [10]. As stress events increase, the stem growth rate is stopped as a tolerance mechanism for the event to minimize carbon starvation and increase hydraulic failure to maximize the probability of survival [11].

Multiple studies have identified that air temperature and soil humidity significantly regulate the stem growth rate [12,13]. Each species is considered to have a range of temperature and water availability for optimal growth, in which photosynthetic processes allow for sufficient carbon (C) capture to cover the maintenance and growth activities of the individual, which is reflected in an increase in secondary growth [14]. Additionally, marginal resistance ranges are available, where the individual regulates photosynthesis and respiration; i.e., growth is suspended and the focus is on surviving [15]. If environmental conditions exceed resistance ranges, the individual will not survive.

The radial growth rate is considered a distinctive trait that identifies individuals susceptible to drought and is thus the first step in characterizing populations and species [7]. Another advantage is being able to reduce the effect of the additive variability associated with individual characteristics (e.g., age, size, competition) and species (e.g., genetics and phenotypic plasticity) [16,17]. Previous studies by Enquist et al. [18] and Fichtler et al. [19] suggested that as the size and age of the tree increases, the stem growth rate decreases, detailing that seasonal growth patterns are maintained, which may be related to environmental variables. Additionally, it is considered that species susceptible to climatic conditions have a low wood density due to vessels with large diameters transporting a high rate of water canopy; in contrast, species with low climatic susceptibility were reported to have a high wood density with smaller vessels as a strategy for resisting cavitation due to water stress [20].

In the case of the tropical region, it is considered an area highly vulnerable to the Niño phenomenon and climate change—specifically, to a reduction in precipitation patterns and an increase in the quantity and magnitude of heat waves, which will cause water stress that can negatively affect the survival and biodiversity of forests [21,22]. For example, Gould et al. [23] mentioned that the latest climate prediction models for 2100, with scenarios stemming from a global temperature increase of 1.5 to 2 °C, indicate a 70% increase in water stress in regions with forest cover in the tropics, and 30% of the area is considered highly susceptible to biodiversity degradation. This point is critical due to the environmental and social role of forests in the tropics, hence the importance of knowing the susceptibility of species growth and survival to climatic conditions and, from there, developing strategies and policies that allow for their conservation in the short and medium term [24,25].

Costa Rica has about 6.5% of the world's diversity [12] and has reported more than 2200 tree species [26]. However, we have robust information on phenology, physiology, and susceptibility to climatic variations for fewer than 10% of tree species [18]. Therefore, this information gap can limit the development of conservation programs for tropical forests and species threatened by climate phenomena [19]. This work aimed to evaluate the interannual stem growth of eight wet tropical forest tree species. We expect to be able to contribute models that relate stem growth to aspects such as wood and climate variables, which will be a first step towards improving our prediction models for the growth and development of species vulnerable to climate change.

## 2. Materials and Methods

### 2.1. Study Site and Species Selection

The study was conducted in an undisturbed primary wet forest in the Cangreja National Park (9.69° N, 84.36° W) (Figure 1a), San José, Costa Rica. The site's climate is type *Am* (Tropical monsoon) according to the Köppen–Geiger classification [27]. With annual temperature values (T) of 23.8 to 29.9 °C and relative humidity (RH) between 62 and 91%, the average photosynthetic active light (PAR) is 1280.3 $\mu$mol m$^{-2}$ s$^{-1}$ (Figure 1b). Finally, the vapor pressure deficit varies from 0.29 to 1.33 kPa, with an average annual precipitation of 3900 mm with events throughout the year (Figure 1c). Regarding the topography, the site shows slopes between 10 and 45°, with clay loam soil (52% clays, 30% silts, and 18% sand), an average pH (depth of 30 cm) of 5.2, and high concentrations of iron and aluminum and deficiencies of potassium and sodium.

Based on data from the permanent plot network of the Tecnológico de Costa Rica, this study considered 48 trees of eight species (6 trees per species) distributed in three permanent plots ($n$ = 2 trees $\times$ species $\times$ plot) (Table 1). The selection criteria considered their abundance, frequency, and dominance within the forest, as well as the presence of individuals in all diameter classes from 10 cm in diameter with more than ten years of information on their growth before the study. Another selection criterion consisted of wood density, with representative species of softwood (lower density of 0.50 g cm$^{-3}$), semi-hardwood (from 0.51 to 0.65 g cm$^{-3}$), and hardwood (greater than 0.66 g cm$^{-3}$) (details in Link et al. [28]). All monitored individuals had their botanical identification

validated by taxonomy professionals from the National Museum of Costa Rica; in addition, all individuals had a diameter at breast height (DBH) greater than 15 cm, without pathogens and pests (Table 1).

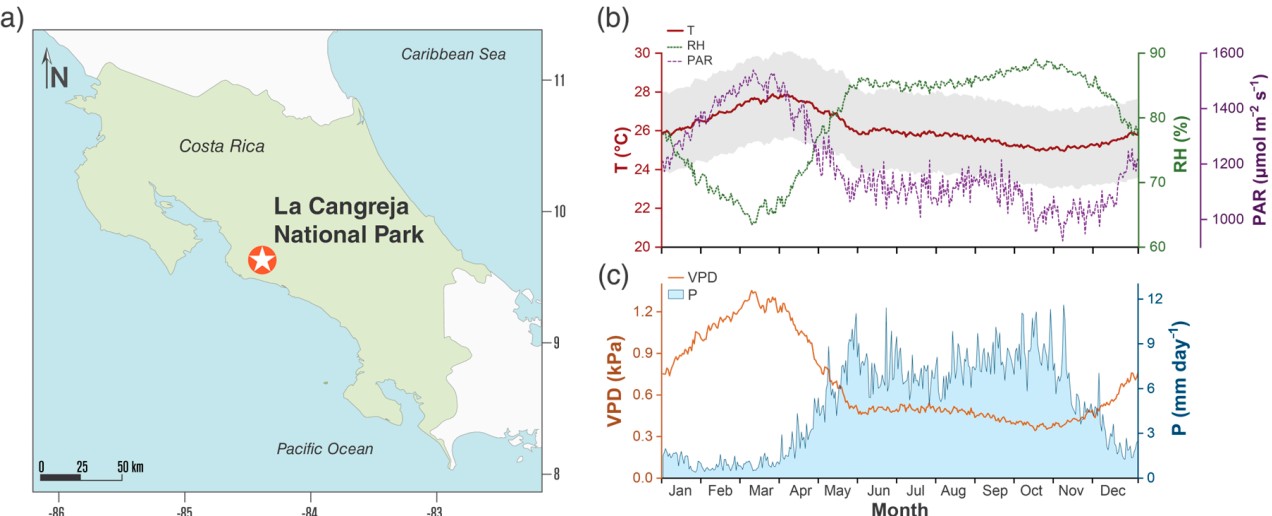

**Figure 1.** Study site location (**a**); average annual values of temperature (T), relative humidity (RH), and photosynthetic active radiation (PAR) (**b**); vapor pressure deficit (VPD) and precipitation (P) (**c**), reported between 1985 to 2015 in the Cangreja National Park. Grey area indicated daily maximum and minimum temperature range.

**Table 1.** Species selected for stem growth measurement of a tropical wet forest located on the Central Pacific slope of Costa Rica.

| Family | Species | Type | DBH (cm) | Bark (mm) | H (m) |
|---|---|---|---|---|---|
| Clusiaceae | *Garcinia madruno* (Kunth) Hammel | Perennial | 21.23 (1.55) | 15.6 (8.8) | 21.39 (2.61) |
| Icacinaceae | *Calatola costaricensis* Standl | Perennial | 19.30 (0.69) | 10.2 (6.9) | 16.61 (1.24) |
| Lecythidaceae | *Gustavia brachycarpa* Pittier | Perennial | 23.19 (1.59) | 4.9 (3.2) | 23.00 (1.90) |
| Lecythidaceae | *Lecythis mesophylla* S.A. Mori | Perennial | 51.82 (9.49) | 20.1 (5.2) | 37.96 (3.52) |
| Malvaceae | *Goethalsia meiantha* (Donn. Sm.) Burret | Deciduous | 26.53 (1.53) | 4.0 (1.6) | 22.75 (2.85) |
| Meliaceae | *Carapa nicaraguensis* C. DC. | Perennial | 37.58 (3.37) | 20.9 (5.8) | 29.45 (2.16) |
| Myristicaceae | *Virola sebifera* Aubl. | Perennial | 25.96 (2.75) | 11.4 (3.9) | 23.45 (2.13) |
| Vochysiaceae | *Vochysia megalophylla* Stafleu | Deciduous | 75.45 (2.69) | 19.2 (7.3) | 37.34 (2.34) |

Values in parentheses represent the standard error. DBH is the diameter at breast height in cm, and H is the total height in m.

### 2.2. Growth Measurement

For the selected individuals, on May 2015, we measured DBH, bark thickness (dendrometer placement), and total height (Table 1). In the case of trees that showed branches, deformities, or bulges, the dendrometer was placed 30 cm above the end point of the irregularity. Stem growth was measured with manual Astralon®-brand dendrometers, highlighting the plastic material's low friction and thermal expansion (coefficient of friction 0.5 in the dry crust and thermal elongation of $7.5 \times 10^{-6}$ K$^{-1}$). The dendrometer was adjusted according to the circumference of each tree, with an excess band of 90 mm. Additionally, bark thickness was measured seasonally with a Suunto tree bark gauge, considering three measurement points per individual tree at the same height as the dendrometer placement. Then, the bimonthly measurement stem increase ($\Delta si$) was estimated with Equation (1), and the annual stem increment (*SI*) was calculated with Equation (2); the bark thickness was previously subtracted in both estimates.

$$\Delta si = si_{i+1} - si_i, \tag{1}$$

$$SI = \sum \Delta si_j, \tag{2}$$

where $\Delta si$ the increase in growth in the monitored period, the increase in stem growth in the monitored period, $si_{i+1}$ is the stem diameter measured at $i + 1$ moment, $si_i$ is the stem diameter measured at $i$ moment, and $SI$ is the summary of stem growth of year $j$; all measurements are in mm and excluding the bark thickness.

After two months of installation (the period considered for adaptation of the dendrometer to site environmental conditions), diametric changes were monitored bimonthly (every 50–60 days). Considering the study period from July 2015 to June 2018 (3 years), manual manipulation of the dendrometers was avoided, except for adjustments due to falling branches, bark, or mechanical failure of the device.

### 2.3. Wood Density Sampling

Wood density was evaluated according to ASTM D2395-17 [29]; a 60 mm-long wood sample was extracted with a Haglöf Auger of 5 mm diameter in each monitored tree. The sample was weighed and sized under green conditions and then dried at 105 °C for 72 h to estimate its dry weight, thus estimating the density with the Archimedes method.

### 2.4. Weather Measurement

The environmental variables of air temperature (T), relative humidity (RH), precipitation (P), and photosynthetic active radiation (PAR) were measured during the study's monitoring period (3 years). T and RH were determined at 15-min intervals with an *ibutton* mini station (Maxim Co., Lawrenceburg, KY, USA). The *ibuttons* were placed at a distance of 500 m from the permanent plots in an area free of vegetation cover that would affect the monitoring ($n$ = three *ibuttons*); then, the vapor pressure deficit (VPD) was calculated via the Penman–Monteith equation [30]. In the case of P, it was carried out at weekly intervals, and three manual rain gauges with a diameter of 15 cm were placed in the same area of the *ibuttons*. Finally, PAR data collected were accessed at a meteorological station 10 km from the study site.

### 2.5. Statistic Analysis

The selection of individuals and the development of the experiment used a simple randomized experimental design. Species in the study period estimated the annual stem growth rate, and then, an analysis of variance (one-way ANOVA) was used to analyze whether the year, tree size, and species influenced differences in the growth rates; if differences were obtained, Tukey's test was used to identify distinctive treatments. Subsequently, the relationship between the annual stem growth rate and wood density was evaluated, for which a Pearson correlation analysis was used, and in cases where the correlation was significant, it was adjusted with a logarithmic model.

A multivariate regression equation was used for the relationship between stem growth rate and climatic variables. To prevent the models from overfitting, we use the two independent variables with higher correlations with stem growth, and the adjusted determination coefficient ($R^2$ adj), root mean square error (RMSE), and mean absolute percentage error (MAPE) per model were calculated. All analyses are considered to have a significance of 0.05 and were performed in R [31].

## 3. Results

### 3.1. Climatic Conditions in the Study Period

The study period presented meteorological conditions different from the historical data for the specific study site, with an increase of 5.1% in the annual values of average T, reaching 27.9 °C (Figure 2a). Maximum T reached 32.3 °C (7.2% increase), and minimum T reached 24.1 °C (1.9% increase); in turn, VPD showed an increase between 1.5 and 6.7%, obtaining average annual values of 1.56 kPa (Figure 2b). For its part, P showed a reduction of 22.5%, with an average annual value of 3020 mm (Figure 2b). Only the RH and PAR variables did not show significant differences compared to historical data (Figure 2).

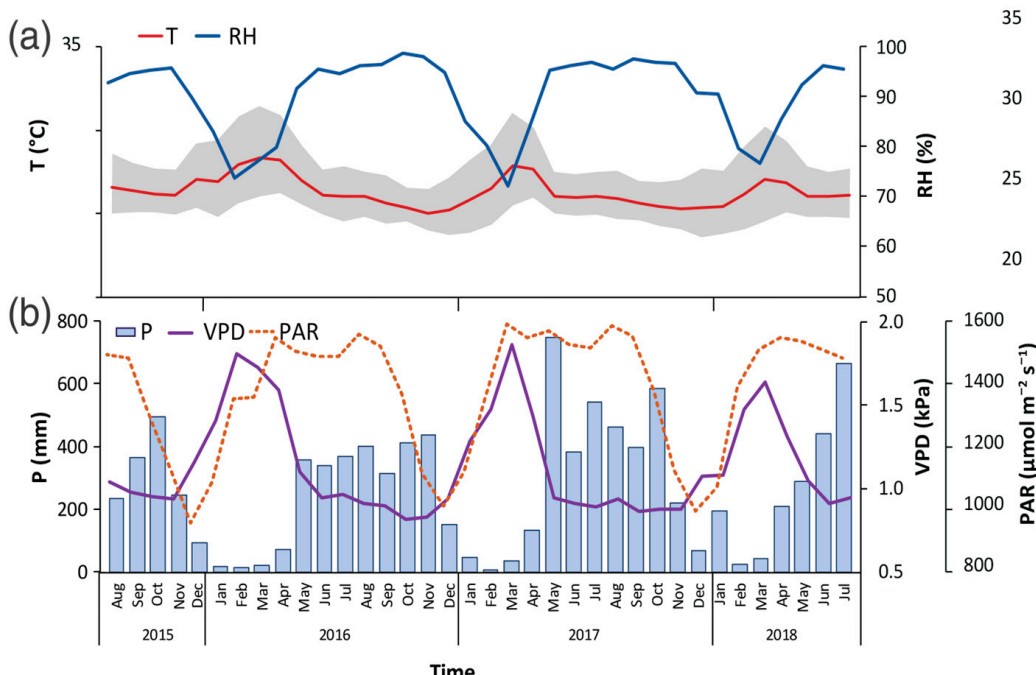

**Figure 2.** Variations of air temperature (T) and relative humidity (RH) (**a**), and precipitation (P), photosynthetic active radiation (PAR), and vapor pressure deficit (VPD) (**b**), measured between 2015 and 2018 in a study site located in the Central Pacific slope of Costa Rica. Grey area indicated daily maximum and minimum temperature range.

*3.2. Stem Growth and Wood Density Partners*

The analysis of variance only showed differences in the variable species (Table 2). The annual stem growth rate showed 0.45 to 4.35 mm yr$^{-1}$ values, and the wood density varied from 0.32 to 0.79 g cm$^{-3}$ (Figure 2). When examining how the stem growth rate changes over time (Figure 3a), all species exhibited increased growth between May and November, corresponding to the rainy season, accounting for 40–70% of annual growth. Surprisingly, low-wood-density species grew seasonally, obtaining stem growth rates <0.02 mm month$^{-1}$ in the driest months.

**Table 2.** Results of analysis of variance (*p*-values) testing measuring year, tree size, species, and their interaction in terms of stem growth rate and wood density in a tropical wet forest located on the Central Pacific slope of Costa Rica.

| Variable | Tree Size | Year | Species | Tree Size × Year | Tree Size × Species | Year × Species | Year × Tree Size × Species |
|---|---|---|---|---|---|---|---|
| Stem growth rate | 0.11 *ns* | 0.30 *ns* | <0.01 * | 0.12 *ns* | 0.44 *ns* | 0.91 *ns* | 0.92 *ns* |
| Wood density | 0.09 *ns* | 0.29 *ns* | <0.01 * | 0.22 *ns* | 0.23 *ns* | 0.82 *ns* | 0.95 *ns* |

\* indicates the significant values at *p* < 0.05; *ns* indicates non-significant values at *p* < 0.05.

Three groups of species were identified: (i) fast-growing species with low-wood density, made up of *V. sebifera*, *V. megalophylla*, and *C. nicaraguensis*, that showed growth rates >3.1 mm yr$^{-1}$ but wood densities < 0.48 g cm$^{-3}$; (ii) moderate-growth and wood-density species comprising *C. costaricensis*, *G. meiantha*, and *G. madruno*, with average stem growth of 1.95 mm yr$^{-1}$ and wood density of 0.54 g cm$^{-3}$; (iii) slow-growing species with high-wood density, composed of *L. mesophylla* and *G. brachycarpa*, with growth rates <0.8 mm yr$^{-1}$ and wood density > 0.65 g cm$^{-3}$. The results indicated that the annual stem growth rate and the species' wood density have an exponential relationship (Figure 3b), indicating that species with high growth rates also have low wood densities. In contrast, species with lower growth rates or seasonal growth patterns have lower wood densities.

This aspect is essential given that fast-growing species showed seasonality in growth compared to high-density wood species that showed growth patterns which were less susceptible to the year's season (Figure 2a).

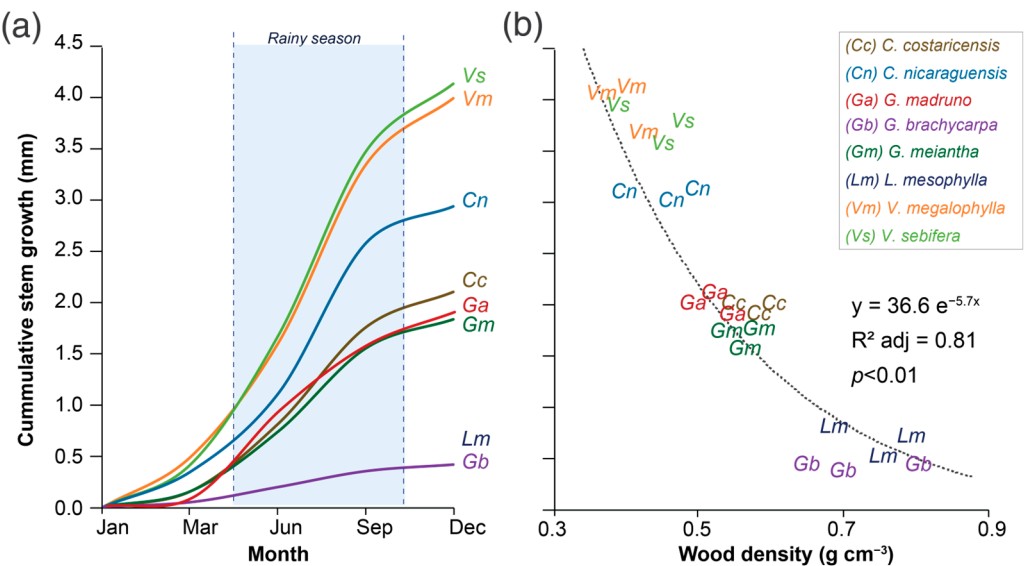

**Figure 3.** Average annual cumulative stem growth rate (**a**) and the relationships between stem growth rate and wood density (**b**) of eight tree species in a tropical wet forest located on the Central Pacific slope of Costa Rica.

### *3.3. Stem Growth Rate Model Adjustment with Climatic Variables*

Pearson's correlation analysis identified that the mean T and P variables significantly correlate to the stem growth rate with R-values > 0.6 (Table 3). In this case, the correlation with mean T was of an inverse linear type (higher T produces lower stem growth rate); whereas with mean P, it was positive linear (higher mean P produces higher stem growth rate). Although the maximum and minimum T showed relationships with the stem growth rate (R-values from 0.45 to 0.60), they were discarded due to their multicollinearity with the average temperature, which may influence the model overfitting. In contrast, minimum P, maximum P, PAR, and RH did not show a significant correlation (R-values from 0.09 to 0.56), so their use in the adjusted model will not contribute to the predictive capacity and the variance explanation. Finally, VPD was rejected because it was a variable that only showed significance in *V. megalophylla*; however, in the rest of the species, it was not significant, with R-values < 0.21.

The adjusted models were completely significant (*p*-values < 0.05) (Table 4), with $R^2$ adj > 0.60 and RMSE < 1.50. The species *C. nicaraguensis*, *G. meiantha*, and *G. brachycarpa* showed higher adjustment ($R^2$ adj > 0.75), with minimum error (RMSE < 0.75) and an average MAPE in stem growth rate of 0.03 mm $yr^{-1}$. In contrast, *C. costaricensis* and *G. madruno* reported lower fitting ($R^2$ adj < 0.70), with maximum error values (average 1.09) and MAPE values between 0.12 and 0.14 mm $yr^{-1}$. Finally, *L. mesophylla*, *V. sebifera*, and *V. megalophylla* showed moderate adjustments between 70% and 72%, with errors lower than 1.00 and MAPE values from 0.06 to 0.07 mm $yr^{-1}$.

**Table 3.** Pearson correlation coefficients between the stem growth rate and climatic variables for eight species in a tropical wet forest located on the Central Pacific slope of Costa Rica.

| Species | Climatic Variable | | | | | | | | |
|---|---|---|---|---|---|---|---|---|---|
| | P (mm) | | | T (°C) | | | Mean PAR ($\mu$mol m$^{-2}$ s$^{-1}$) | Mean RH (%) | Mean VPD (kPa) |
| | Minimum | Mean | Maximum | Minimum | Mean | Maximum | | | |
| *C. costaricensis* | 0.552 *ns* | 0.732 ** | 0.233 *ns* | −0.588 * | −0.696 * | −0.516 * | 0.011 *ns* | 0.011 *ns* | 0.201 *ns* |
| *C. nicaraguensis* | 0.366 *ns* | 0.659 ** | 0.411 *ns* | −0.536 * | −0.756 * | −0.486 * | 0.095 *ns* | 0.028 *ns* | −0.200 *ns* |
| *G. madruno* | 0.220 *ns* | 0.601 ** | 0.332 *ns* | −0.613 * | −0.803 * | −0.537 * | 0.010 *ns* | 0.009 *ns* | 0.009 *ns* |
| *G. meiantha* | 0.422 *ns* | 0.705 ** | 0.150 *ns* | −0.520 * | −0.722 * | −0.499 * | 0.013 *ns* | 0.055 *ns* | −0.305 *ns* |
| *G. brachycarpa* | 0.332 *ns* | 0.687 ** | 0.144 *ns* | −0.557 * | −0.677 * | −0.600 * | 0.021 *ns* | 0.045 *ns* | 0.045 *ns* |
| *L. mesophylla* | 0.255 *ns* | 0.603 ** | 0.355 *ns* | −0.605 * | −0.775 * | −0.511 * | 0.009 *ns* | 0.008 *ns* | 0.180 *ns* |
| *V. sebifera* | 0.310 *ns* | 0.680 ** | 0.255 *ns* | −0.597 * | −0.777 * | −0.509 * | 0.012 *ns* | 0.029 *ns* | 0.029 *ns* |
| *V. megalophylla* | 0.250 *ns* | 0.641 ** | 0.366 *ns* | −0.533 * | −0.703 * | −0.544 * | 0.015 *ns* | 0.009 *ns* | 0.390 * |

P is the precipitation, T is air temperature, PAR is the photosynthetic active radiation, RH is the relative humidity, and VPD is the vapor pressure deficit. *ns* indicates non-significant values at $p < 0.05$; * Indicates the significant values at $p < 0.05$; and ** indicates the significant values at $p < 0.01$.

**Table 4.** Multivariate models adjusted the stem growth rate based on climatic variables per species in a tropical wet forest located on the Central Pacific slope of Costa Rica.

| Species | Regression Model | R$^2$ adj | RMSE | *p*-Value | MAPE |
|---|---|---|---|---|---|
| *C. costaricensis* | Sg = 1.289 + 0.001 × P − 0.053 × T | 0.68 | 1.00 | 0.013 * | 0.14 |
| *C. nicaraguensis* | Sg = 3.849 + 0.001 × P − 0.092 × T | 0.79 | 0.61 | 0.011 * | 0.04 |
| *G. madruno* | Sg = 1.463 + 0.001 × P − 0.036 × T | 0.62 | 1.18 | 0.001 ** | 0.12 |
| *G. meiantha* | Sg = 1.929 + 0.001 × P − 0.069 × T | 0.74 | 0.78 | 0.001 ** | 0.03 |
| *G. brachycarpa* | Sg = 0.206 + 0.001 × P − 0.005 × T | 0.78 | 0.73 | 0.001 ** | 0.02 |
| *L. mesophylla* | Sg = 0.197 + 0.001 × P − 0.005 × T | 0.72 | 0.71 | 0.002 ** | 0.07 |
| *V. sebifera* | Sg = 3.569 + 0.001 × P − 0.132 × T | 0.70 | 0.92 | 0.001 ** | 0.06 |
| *V. megalophylla* | Sg = 3.078 + 0.002 × P − 0.485 × T | 0.72 | 0.99 | 0.001 ** | 0.07 |

Sg is the stem growth rate, T is the mean air temperature, P is the precipitation, R$^2$ adj is the adjusted determination coefficient, RMSE is the root mean square error, and MAPE is the mean absolute percentage error. * indicates the significant values at $p < 0.05$; ** indicates the significant values at $p < 0.01$.

## 4. Discussion

### 4.1. Dynamic Stem Growth Rate

According to our findings, the rainy season, with higher amounts of precipitation and colder temperatures, had the highest stem growth rates for single-track species (Figure 2). This pattern is similar to that reported by Wang and Hamzah [32] for two pioneer species in tropical forests; rainy season conditions in the tropics reduce water stress in forests, the air temperature is reduced, and water capture by the root system is facilitated, influencing higher rates of photosynthetic assimilation and C fixation. Other studies, such as that of Anderson et al. [33], indicated that this is a period in which VPD is reduced, which causes a lower rate of transpiration in the canopy, increasing the efficiency of photosynthetic processes. This aspect is vital in fast-growing pioneer species that consume high water rates and are susceptible to high temperatures [34].

Oberhuber and Gruber [35] found that tropical forests in drought conditions showed stagnation in the growth and an increase in the mortality of fast-growing species, with stomatal regulation and the reduction of leaf area being fundamental strategies with which to survive adverse abiotic events. In the same way, Iberle et al. [36] posited that wet tropical forest species show less resistance to drought because they have evolutionarily established themselves in environments with an abundance of water resources, which allowed them to maintain high transpiration rates despite high temperatures in the environment; however, in drought conditions, they tend to have greater vulnerability, attributed to anatomical traits at the leaf level (e.g., density and size of stomata).

The seasonality of the stem growth rate is an interesting variable in analyzing ecosystems' resilience to climate change [28,37]. In tropical dry forests, it has been reported that species with low density and high growth rates tend to have a low tolerance to

drought; trees present a reduced regulation of transpiration and the loss-of-leaf-area mechanism, in addition to showing xylem anatomy susceptible to hydraulic failure due to embolism [12,38]. However, the growth of forest species must be carefully analyzed, along with the additive effects associated with the individual (e.g., size and age) or tree competition [1,39], in order that growth patterns can be identified at the population level, which is the first step towards discriminating and identifying vulnerable and drought-resistant species [40].

### 4.2. Role of Wood Density in Stem Growth Rate Stationary

In this study, a decrease in wood density was determined as the annual growth of the individual increased (Figure 3). The most significant relationship was found in low-density species, considered softwoods. Said behavior is similar to that explained by Li et al. [37] for species from the south of France. A linear model was developed between growth and wood density, determining that as growth increased, the density decreased, with the rainy season being the season that influenced growth the most. Low-density wood tended to have larger vessels and porous cell structures due to the constant cell division in the cambium, which tends to form many cells with small cell walls and significant size. The opposite occurs with species with high densities; there is a smaller cambium development process due to greater wall thickness and smaller cell diameter [7].

Cardil et al. [17] found that species with slow growth and high wood density presented greater adaptability to conditions of heat stress compared to species with low wood density due to their susceptibility to embolism because the xylem anatomy has a low density of vessels but a large diameter, which means that the individual must invest high amounts of non-structural carbohydrates (NSC) into their elimination. Similarly, Wang and Hamzah [32] posited that species with high wood density presented high water use efficiency; they have greater regulation of photosynthesis and respiration, in addition to their anatomical structure, allowing them to withstand drought events and maintain continuous growth rates. Finally, Jiménez et al. [41] considered high-wood-density species that showed a low growth rate as a strategy for efficient energy and water use, benefiting the accumulation of NSC.

### 4.3. Equations for Estimating Growth as a Function of Environmental Variables

Developing growth models with environmental variables is essential to evaluate potential effects on mortality and the dynamics of species, particularly endemic species or those with reduced populations that present extensive phenological periods [7]. Hu and Fan [42] posited that having growth prediction models that consider climatic variables allows one to analyze the susceptibility of tropical species to extreme climatic events and, in turn, what survival responses and growth rates can be considered acceptable or critical for conservation. Additionally, Campoe et al. [3] employed this model to predict the effect of climate on species' productivity and harvest cycles under drought conditions, which must impact the production of timber forest resources that involve society at large and communities close to these ecosystems. Likewise, Clark et al. [13] suggested that this model is vital to regulating protected species in forestry utilization plans and defining harvest periods to reduce forest biodiversity degradation.

Aldea et al. [43] highlighted that as climate-based growth model development increases, decision-making and legislation tools will be available to protect the species most susceptible to climate change. As evidenced in this study, the species' stem growth and development behaviors varied significantly depending on the density of the wood. In the process of protecting and conserving arboreal species, not only do the size and distribution of a species constitute the criteria for its conservation or legal protection, so too do its growth rate and resistance capacity to drought and heat stress [44].

## 5. Conclusions

It was identified that the rainy season months, with their greater availability of water and lower annual daily temperature, yielded the highest growth rates. Also, a difference was found in the annual stem growth trend between species, with species with lower wood density showing seasonality in growth and susceptibility to dry periods; in contrast, species with high density did not show seasonality in growth, an aspect that suggests they have adaptation mechanisms to climatic conditions that allow for their growth. Precipitation and average air temperature were the environmental variables that most influenced the stem growth rate. This aspect is critical to beginning the characterization of wet tropical forest species' resistance to the drought and heat stress associated with climate change and determining which species should be included in biodiversity conservation programs in the short and medium term.

**Author Contributions:** J.C.V., D.A.-A. and M.C.-U. designed and performed the research framework, collected and analyzed the data, and wrote and prepared the original draft. J.C.V. and D.A.-A. designed and supervised the study, reviewed the manuscript, and approved the final draft. N.Z.-V. participated in species identification and reviewed the manuscript draft. All authors have read and agreed to the published version of the manuscript.

**Funding:** This work was funded by Virerectoría de Ivestigación y Extención (VIE) del Tecnológico de Costa Rica (ITCR) in the project No. 1401069: "*Estudios sobre aspectos hidráulicos de árboles en el trópico americano y su efecto en el crecimiento*".

**Data Availability Statement:** Data are contained within the article; further inquiries can be directed to the corresponding author.

**Acknowledgments:** We gratefully acknowledge the support of many Tecnologíco de Costa Rica professionals, the technical department of National System of Conservation Areas (SINAC) and ANID agency via DOCTORADO BECAS CHILE/2020-21202023.

**Conflicts of Interest:** The authors declare no conflicts of interest.

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
