# Peer review of "Influence of Climatic Variables on the Stem Growth Rate in Trees of a Tropical Wet Forest"

_conservation, doi:10.3390/conservation4020010_

Round 1
Reviewer 1 Report
Comments and Suggestions for Authors
Dear Authors,
The submitted manuscript is well-written, the experiment is well-organized, and answers the research questions. In my view this paper should be published in the MDPI Forests journal, instead of this journal. EiC can be transferred to this paper in the suggested journal because this is a typical forestry paper. I can not see some bigger problems and language is on publishable levels. Reconsider my suggestion about the change of journal and good luck with publishing this paper.
Comments on the Quality of English Language...
Author Response
We appreciate the comments provided, we have improved drafting points of the manuscript. The editor has mentioned to us that it is a topic related to the magazine, so we have decided to continue the process with Conservation.
Reviewer 2 Report
Comments and Suggestions for Authors
Dear authors,
The article is well written. However, there are some small questions and corrections to be made.
Attached document.

Author Response
Q1. Table 1: The first time the species appears, the authority (Garcinia madruno (Kunth) Hammel) must be indicated. Do this for all species.
We appreciate this important observation, we have added the authorship in all the scientific names of the species in the study.
Q2. Table 1: What is this? error or standard deviation
The term in parentheses is the standard error. We have added the detail at the bottom of table 1.
Q3. Line 197: Verify
The correct term is Pearson correlation, all articles that are correct have been reviewed.
Q4. Table 3: Verify
The correct term is Pearson correlation, all articles that are correct have been reviewed.
Q5. Line 305: There is no need to paste this information, it is already in the results and summary.
We appreciate the observation, we have removed the sentence in the conclusion.
Reviewer 3 Report
Comments and Suggestions for Authors
conservation-2875062
My comments:
Good job! This manuscript can be considered for publication after a major revision.
Lines 96-97: Each species has 6 samples, accounting for 6 * 8 = 48 samples, right?
How many plots are the 6 samples distributed in?
Lines 112-113: How to measure bark thickness? Does stem growth rate contain bark thickness? Stem growth rate should be quantified by a formula.
Lines 130-138 (Section: 2.4 Weather measurement): How to measure the climate variables? In times or in space?
Figure 2: There is a little cover in the upper left corner of fig. 2.
Table 3: Why the minimum, maximum and mean values of air temperature were considered, but the precipitation not?
Table 4: It would be better that if tree size and tree species composition could be considered in the independent variables.
Author Response
Q1. Lines 96-97: Each species has 6 samples, accounting for 6 * 8 = 48 samples, right?
The statement is complete, we have improved the wording on line 97: "The study considered 48 trees of eight species (6 trees per species) distributed in three permanent plots (n= 2 trees x species x plot)".
Q2. How many plots are the 6 samples distributed in?
The study considered three permanent plots, as edited in Q1 we have included that detail in Lines 97-98
Q3. Lines 112-113: How to measure bark thickness? Does stem growth rate contain bark thickness? Stem growth rate should be quantified by a formula.
We have included the detail requested on line 119: "Additionally, bark thickness was measured seasonally with a Suunto tree bark gauge, considering three measurement points per individual at the same height as the dendrometer placement"; other details in equation 1 and 2.
Q4. Lines 130-138 (Section: 2.4 Weather measurement): How to measure the climate variables? In times or in space?
We consider an important aspect to improve, we have reformulated the wording of the paragraph on line 134-144 as follows: "The environmental variables of air temperature (T), relative humidity (RH), precipitation (P), and photosynthetic active radiation (PAR) were measured during the study's monitoring period (4 years). T and RH were determined at 15-minute intervals with an ibutton mini station (Maxim Co., USA). The ibuttons were placed at a distance of 500 m from the permanent plots in an area free of vegetation cover that would affect the monitoring (n= three ibuttons), then, the vapor pressure deficit (VPD) was calculated by Penman-Monteith equation [30]. In the case of P was carried out at weekly intervals, three manual rain gauges with a diameter of 15 cm were placed in the same area of the ibuttons. Finally, PAR data collected were accessed at a meteorological station 10 km from the study site."
Q5. Figure 2: There is a little cover in the upper left corner of fig. 2.
Corrected!
Q6. Table 3: Why the minimum, maximum and mean values of air temperature were considered, but the precipitation not?
We appreciate this observation, we had previously discarded the minimum and maximum precipitation as it was not significant, so we have included these values in Table 3.
Q7. Table 4: It would be better that if tree size and tree species composition could be considered in the independent variables.
We appreciate the observation, as shown in Table 2 when including the independent variable of tree size, this did not show significance in the interactions, so we have not made an inclusion within the equations of Table 4 because it is not a significant variable in the study.
Round 2
Reviewer 3 Report
Comments and Suggestions for Authors
Good job! The manuscript can be considered for publication in Conservation.